# DEFUSE: DEBUGGING CLASSIFIERS THROUGH DISTILLING UNRESTRICTED ADVERSARIAL EXAMPLES

## ABSTRACT

With the greater proliferation of machine learning models, the imperative of diagnosing and correcting bugs in models has become increasingly clear. As a route to better discover and fix model bugs, we propose *failure scenarios*: regions on the data manifold that are incorrectly classified by a model. We propose an end-to-end debugging framework called Defuse to use these regions for fixing faulty classifier predictions. The Defuse framework works in three steps. First, Defuse *identifies* many unrestricted adversarial examples—naturally occurring instances that are misclassified—using a generative model. Next, the procedure *distills* the misclassified data using clustering into failure scenarios. Last, the method *corrects* model behavior on the distilled scenarios through an optimization based approach. We illustrate the utility of our framework on a variety of image data sets. We find that Defuse identifies and resolves concerning predictions while maintaining model generalization.

## 1 INTRODUCTION

Debugging machine learning (ML) models is a critical part of the ML development life cycle. Uncovering bugs helps ML developers make important decisions about both development and deployment. In practice, much of debugging uses aggregate test statistics (like those in leader board style challenges [Rajpurkar et al. (2016)]) and continuous evaluation and monitoring post deployment [Liberty et al. (2020), Simon (2019)]. However, additional issues arise with over-reliance on test statistics. For instance, aggregate statistics like held out test accuracy are known to overestimate generalization performance [Recht et al. (2019)]. Further, statistics offer little insight nor remedy for specific model failures [Ribeiro et al. (2020); Wu et al. (2019)]. Last, reactive debugging of failures as they occur in production does little to mitigate harmful user experiences [La Fors et al. (2019)]. Several techniques exist for identifying undesirable behavior in machine learning models. These methods include explanations [Ribeiro et al. (2016); Slack et al. (2020b); Lakkaraju et al. (2019); Lundberg & Lee (2017)], fairness metrics [Feldman et al. (2015), Slack et al. (2020a)], data set replication [Recht et al. (2019); Engstrom et al. (2020)], and behavioral testing tools [Ribeiro et al. (2020)]. However, these techniques do not provide methods to remedy model bugs or require a high level of human supervision. To enable model designers to discover and correct model bugs beyond aggregate test statistics, we analyze *unrestricted adversarial examples*: instances on the data manifold that are misclassified [Song et al. (2018)]. We identify model bugs through diagnosing common patterns in unrestricted adversarial examples.

In this work, we propose Defuse: a technique for debugging classifiers through distilling[1] unrestricted adversarial examples. Defuse works in three steps. First, Defuse *identifies* unrestricted adversarial examples by making small, semantically meaningful changes to input data using a variational autoencoder (VAE). If the classifier prediction deviates from the ground truth label on the altered instance, it returns the data instance as a potential model failure. This method employs similar techniques from [Zhao et al. (2018)]. Namely, small perturbations in the latent space of generative models can produce images that are misclassified. Second, Defuse *distills* the changes through clustering on the unrestricted adversarial example's latent codes. In this way, Defuse diagnoses regions in the latent space that are problematic for the classifier. This method produces a set of

---

[1]We mean distilling in the sense of "to extract the most important aspects of" and do not intend to invoke the knowledge distillation literature [Hinton et al. (2014)].

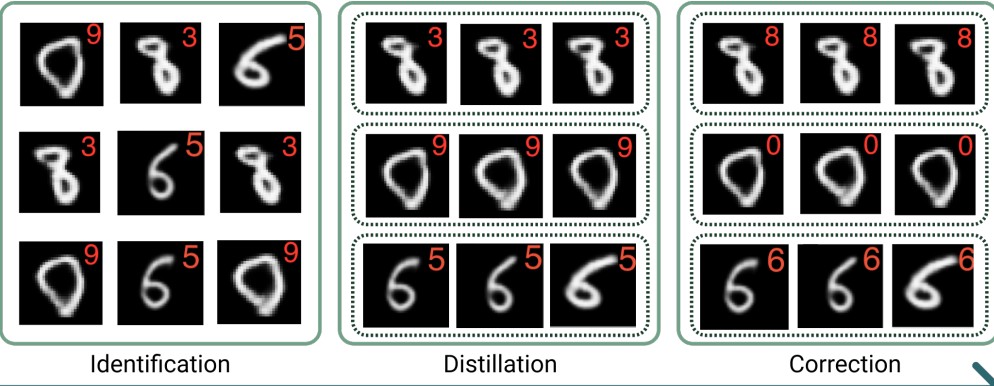

Figure 1: **Running Defuse on a MNIST classifier.** The (handpicked) images are examples from three failure scenarios identified from running Defuse. The red digit in the upper right hand corner of the image is the classifier's prediction. Defuse initially identifies many model failures. Next, it aggregates these failures in the distillation step for annotator labeling. Last, Defuse tunes the classifier so that it correctly classifies the images, with minimal change in classifier performance. Defuse serves as an end-to-end framework to diagnose and debug errors in classifiers.

clusters in the latent space where it is likely to find misclassified data. We call these localities *failure scenarios*. An annotator reviews the failure scenarios and assigns the correct label— one label per scenario. Third, Defuse *corrects* the model behavior on the discovered failure scenarios through optimization. Because we use a generative clustering model to describe the failure scenarios, we sample many unrestricted adversarial examples and finetune to fix the classifier. Critically, failure scenarios are highly useful for model debugging because they reveal high level patterns in the way the model fails. By understanding these consistent trends in model failures, model designers can more effectively understand problematic deployment scenarios for their models.

To illustrate the usefulness of failure scenarios, we run Defuse on a classifier trained on MNIST and provide an overview in figure 1. In the identification step (first pane in figure 1), Defuse generates unrestricted adversarial examples for the model. The red number in the upper right hand corner of the image is the classifier's prediction. Although the classifier achieves high test set performance, we find naturally occurring examples that are classified incorrectly. Next, the method performs the distillation step (second pane in figure 1). The clustering model groups together similar failures for annotator labeling. We see that similar mistakes are grouped together. For instance, Defuse groups together a similar style of incorrectly classified eights in the first row of the second pane in figure 1. Next, Defuse receives annotator labels for each of the clusters.[2] Last, we run the correction step using both the annotator labeled data and the original training data. We see that the model correctly classifies the images (third pane in figure 1). Importantly, the model maintains its predictive performance, scoring 99.1% accuracy after tuning. We see that Defuse enables model designers to both discover and correct naturally occurring model failures.

We provide the necessary background in Defuse (§2). Next, we detail the three steps in Defuse: *identification*, *distillation*, and *correction* (§3). We then demonstrate the usefulness of Defuse on three image data sets: MNIST [LeCun et al. (2010)], the German traffic signs data set [Stallkamp et al. (2011)], and the Street view house numbers data set [Netzer et al. (2011)], and find that Defuse discovers and resolves critical bugs in high performance classifiers trained on these datasets (§4).

## 2 NOTATION AND BACKGROUND

In this section, we establish notation and background on unrestricted adversarial examples. Though unrestricted adversarial examples can be found in many domains, we focus on Defuse applied to image classification.

---

[2]We assign label 8 to the first row in the second pane of figure 1, label 0 to the second row, and label 6 to the third row.

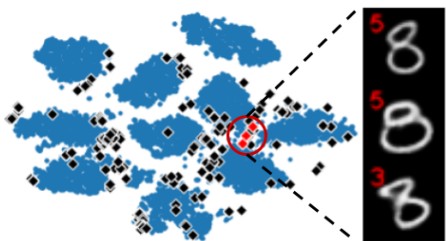

Figure 2: **Providing intuition for failure scenarios** through a t-SNE visualization of the latent space of MNIST. The black diamonds correspond to the latent codes of unrestricted adversarial examples. The blue circles are the latent codes of images from the training set. The images are three decoded latent codes (the red dots), where the red number in the left hand corner is the classifier label. We see that there are regions with higher densities of adversarial examples

**Unrestricted adversarial examples**   Let $f : \mathbb{R}^N \to [0, 1]^C$ denote a classifier that accepts a data point $x \in X$, where $X$ is the set of legitimate images. The classifier $f$ returns the probability that $x$ belongs to class $c \in \{1, ..., C\}$. Next, assume $f$ is trained on a data set $\mathcal{D}$ consisting of $d$ tuples $(x, y)$ containing data point $x$ and ground truth label $y$ using loss function $\mathcal{L}$. Finally, suppose there exists an oracle $o : x \in X \to \{1, ..., C\}$ that outputs a label for $x$. We define unrestricted adversarial examples as the set $\mathcal{A}_N := \{x \in X \mid o(x) \neq f(x)\}$ [Song et al. (2018)].

**Variational Autoencoders (VAEs)**   In order to discover unrestricted adversarial examples, it is necessary to model the set of legitimate images. We use a VAE to create such a model. A VAE is composed of an encoder and a decoder neural networks. These networks are used to model the relationship between data $x$ and latent factors $z \in \mathbb{R}^K$. Where $x$ is generated by some ground truth latent factors $v \in \mathbb{R}^M$, we wish to train a model such that the learned generative factors closely resemble the true factors: $p(x|v) \approx p(x|z)$. In order to train such a model, we employ the $\beta$-VAE [Higgins et al. (2017)]. This technique produces encoder $q_\phi(z|x)$ that maps from the data and latent codes and decoder $p_\theta(x|z)$ that maps from codes to data.

## 3 METHODS

### 3.1 FAILURE SCENARIOS

We begin by formalizing our notion of *failure scenarios*. Let $z \in \mathbb{R}^K$ be the latent codes corresponding to image $x \in X$ and $q_\phi(\cdot) : x \to z$ be the encoder mapping the relationship between images and latent codes.

**Definition 3.1.** Failure scenario. *Given a constant $\epsilon > 0$, vector norm $|| \cdot ||$, and point $z'$, a failure scenario is a set of images $\mathcal{A}_R = \{x \in X \mid \epsilon > ||q_\phi(x) - z'|| \wedge o(x) \neq f(x)\}$.*

Previous works that investigate unrestricted adversarial examples look for specific instances where the oracle and the model disagree [Song et al. (2018); Zhao et al. (2018)]. We instead look for regions in the latent space where this is the case. Because the latent space of the VAE tends to take on Gaussian form due to the prior, we can use euclidean distance to define these regions. If we were to define failure scenarios on the original data manifold, we may need a much more complex distance function. Because it is likely too strict to assume the oracle and model disagree on *every* instance in such a region, we also introduce a relaxation.

**Definition 3.2.** Relaxed failure scenario. *Given a constant $\epsilon > 0$, vector norm $|| \cdot ||$, point $z'$, and threshold $\rho$, a relaxed failure scenario is a set of images $\mathcal{A}_f = \{x \in X \mid \epsilon > ||q_\phi(x) - z'||\}$ such that $|\{x \in \mathcal{A}_f \mid o(x) \neq f(x)\}| / |\mathcal{A}_f| > \rho$.*

In this work, we adopt the latter definition of failure scenarios. To concretize failure scenarios and provide evidence for their existence, we continue our MNIST example from figure 1. We plot the t-SNE embeddings of the latent codes of 10000 images from the training set and 516 unrestricted

adversarial examples created during the identification step in figure 2 (details of how we generate unrestricted adversarial examples in section 3.2.1). We see that the unrestricted adversarial examples are from similar regions in the latent space.

## 3.2 DEFUSE

In this section, we introduce Defuse: our procedure for identifying and correcting classifier performance on failure scenarios. First, we explain how we identity unrestricted adversarial examples using VAEs. Next, we describe our clustering approach that distills these instances into *failure scenarios*. Last, we introduce our approach to correct classifier predictions on the failure scenarios.

### 3.2.1 IDENTIFYING UNRESTRICTED ADVERSARIAL EXAMPLES

This section describes the *identification* step in Defuse (first pane in figure 1). The aim of the *identification* step is to generate many unrestricted adversarial examples. In essence, we encode all the images from the training data. We perturb the latent codes with a small amount of noise drawn from a Beta distribution. We save instances that are classified differently from ground truth by $f$ when decoded. By perturbing the latent codes with a small amount of noise, we expect the decoded instances to have small but semantically meaningful differences from the original instances. Thus, if the classifier prediction deviates on the perturbation the instance is likely misclassified. We denote the set of unrestricted adversarial examples for a single instance $\psi$. We generate unrestricted adversarial examples over each instance $x \in X$ producing a set of unrestricted adversarial $\Psi$ containing the $\psi$ produced for each instance $x$. Pseudo code of the algorithm for generating a single unrestricted adversarial example is given in algorithm 1 in appendix A.

Our technique is related to the method for generating *natural adversarial examples* from [Zhao et al. (2018)] — a very similar but slightly different concept from unrestricted adversarial examples. The authors use a similar stochastic search method in the latent space of a GAN. They start with a small amount of noise and increase magnitude of the noise until they find a unrestricted adversarial example. Thus, they save only the unrestricted adversarial examples which are minimally distant from a data point. They also save images that differ in prediction from the original decoded instance. Because we iterate over the entire data set, it is simpler to keep the level of noise fixed and sample a predetermined number of times. In addition, we save images that differ in ground truth label from the original decoded instance because we seek to debug a classifier. Meaning, if the original instance is misclassified we wish to save this instance as a model failure.

### 3.2.2 DISTILLING FAILURE SCENARIOS

This section describes the *distillation* step in defuse (second pane of figure 1). The goal of the distillation step is to cluster the latent codes of the set of unrestricted adversarial examples $\Psi$ in order to diagnose failure scenarios. We require our clustering method to (1) infer the correct number of clusters from the data and (2) be capable of generating instances of each cluster. We need to infer the number of clusters from the data because the number of failure scenarios are unknown ahead of time. Further, we must be capable of generating many instances from each cluster so that we have enough data to finetune on in order to correct the faulty model behavior. In addition, generating many failure instances enables model designers to see numerous examples from the failure scenarios, which encourages understanding of the model failure modes. Though any such clustering method that fits this description could be used for distillation, we use a Gaussian mixture model (GMM) with Dirichlet process prior. We use the Dirichlet process because it nicely describes the clustering problem where the number of mixtures is unknown before hand, fulfilling our first criteria [Sudderth (2006)]. Additionally, because the model is generative, we can sample new instances, which satisfies our second criteria.

In pratice, we use the truncated stick breaking construction of the dirchlet process, where $K$ is the upper bound of the number of mixtures. The truncated stick breaking construction simplifies inference making computation more efficient [Sudderth (2006)]. The method outputs a set of clusters $\theta_j = (\mu_j, \sigma_j, \pi_j)$ where $j \in \{1, ..., K\}$. The parameters $\mu$ and $\sigma$ describe the mean and variance of a multivariate normal distribution and $\pi$ indicates the cluster weight. To perform inference on the model, we employ expectation maximization (EM) described in [Bishop (2006)] and use the implementation provided in [Pedregosa et al. (2011)]. Once we run EM and determine the parameter

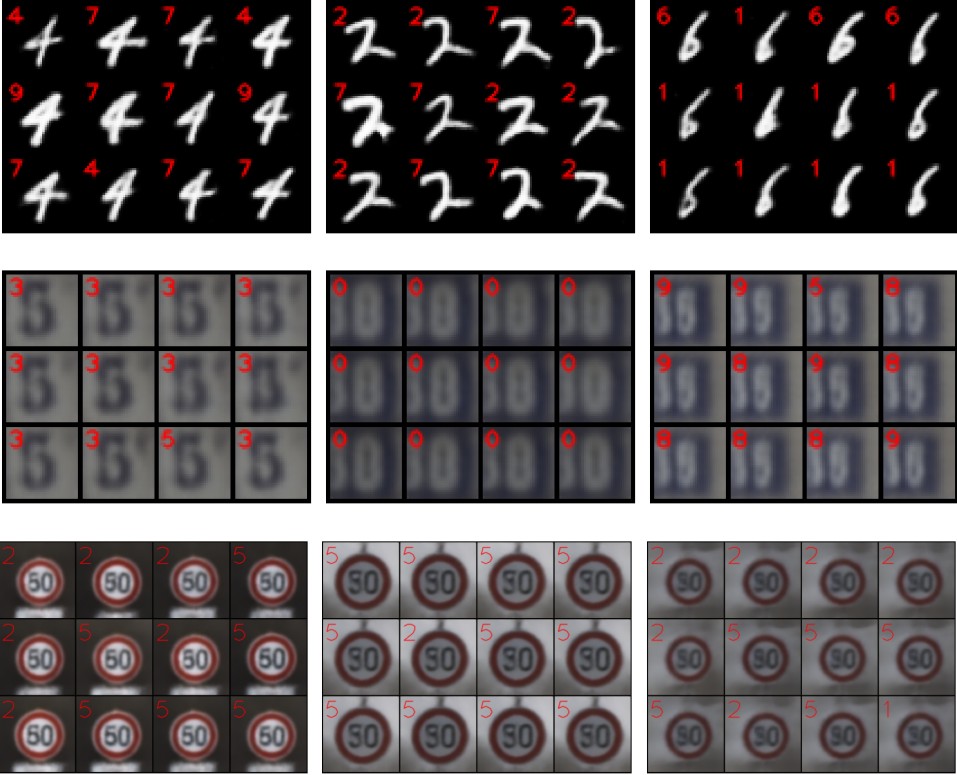

Figure 3: **Samples from three failure scenarios** from each dataset. **First row:** The MNIST failure scenarios. These scenarios were labeled 4, 2, 6 in order from left to right. **Second row:** The SVHN failure scenarios labeled 5, 8, and 5 from left to right. **Third row:** The German signs failure scenarios. The label 1 corresponds to 30km/h, 2 to 50km/h, and 5 to 80km/h. The first and second were labeled 2 while the third was labeled 1. Defuse finds significant bugs in the classifiers.

values, we throw away cluster components that are not used by the model. We fix some small $\epsilon$ and define the set of failure scenarios $\Lambda$ generated at the distillation step as: $\Lambda := \{(\mu_j, \Sigma_j, \pi_j) | \pi_j > \epsilon\}$.

### 3.2.3 CORRECTING FAILURE SCENARIOS

**Labeling** First, an annotator assigns the correct label to the failure scenarios. For each failure scenario identified in $\Lambda$, we sample $Q$ latent codes from $z \sim \mathcal{N}(\mu_j, \tau \cdot \sigma_j)$. Here, $\tau \in \mathbb{R}$ is a hyperparameter that controls the diversity of samples from the failure scenario. Because it could be possible for multiple ground truth classes to be present in a failure scenario, we set this parameter tight enough such that the sampled instances are from the same class to make labeling easier. We reconstruct the latent codes using the decoder $p_\theta(x|z)$. Next, an annotator reviews the reconstructed instances from the scenario and decides whether the scenario constitutes a model failure. If so, the annotator assigns the correct label to all of the instances. The correct label constitutes a single label for all of the instances generated from the scenario. We repeat this process for each of the scenarios identified in $\Lambda$ and produce a dataset of failure instances $\mathcal{D}_f$. Pseudo code for the procedure is given in algorithm 2 in appendix A.

**Finetuning** We finetune on the training data with an additional regularization term to fix the classifier performance on the failure scenarios. The regularization term is the cross entropy loss between the identified failure scenarios and the annotator label. Where $CE$ is the cross entropy loss applied to the failure instances $\mathcal{D}_f$ and $\lambda$ is the hyperparameter for the regularization term, we optimize the following objective using gradient descent: $\mathcal{F}(\mathcal{D}, \mathcal{D}_f) = \mathcal{L}(\mathcal{D}) + \lambda \cdot CE(\mathcal{D}_f)$. This objective encourages the model to maintain its predictive performance on the original training data while en-

couraging the model to predict the failure instances correctly. The regularization term $\lambda$ controls the pressure applied to the model to classify the failure instances correctly.

## 4 EXPERIMENTS

### 4.1 SETUP

**Datasets** We evaluate Defuse on three datasets: MNIST [LeCun et al. (2010)], the German Traffic Signs dataset [Stallkamp et al. (2011)], and the Street view house numbers dataset (SVHN) [Netzer et al. (2011)]. MNIST consists of $60,000$ 32X32 handwritten digits for training and $10,000$ digits for testing. The images are labeled corresponding to the digits $0 - 9$. The German traffic signs data set includes $26,640$ training and $12,630$ testing images of size 128X128. We randomly split the testing data in half to produce a validation and testing set. The images are labeled from $43$ different classes to indicate the type of traffic signs. The SVHN data set consists of $73,257$ training and $26,032$ testing images of size 32X32. The images include digits of house numbers from Google streetview with labels $0 - 9$. We split the testing set in half to produce a validation and testing set.

**Models** On MNIST, we train a CNN scoring $98.3\%$ test set accuracy following the architecture from [Paszke et al. (2019)]. On German traffic signs and SVHN, we finetune a Resnet18 model pretrained on ImageNet [He et al. (2016)]. The German signs and SVHM models score $98.7\%$ and $93.2\%$ test accuracy respectively. We train a $\beta$-VAE on all available data from each data set to model the set of legitimate images in Defuse. We use an Amazon EC2 P3 instance with a single NVIDIA Tesla V100 GPU for training. We follow similar architectures to [Higgins et al. (2017)]. We set the size of the latent dimension $z$ to 10 for MNIST/SVHN and 15 for German signs. We provide our $\beta$-VAE architectures in appendix B.

**Defuse** In the identification step, we fix the parameters of the Beta distribution noise $a$ and $b$ to $a = b = 50.0$ for MNIST and $a = b = 75.0$ for SVHN and German signs. We found these parameters were good choices because they produce a very small amount of perturbation noise making the decoded instance only slightly different than the original instance. During distillation, we set the upper bound on the number of components $K$ to $100$. We generally found the actual number of clusters to be much lower than this level. Thus, this serves as an appropriate upper bound. We also fixed the weight threshold for clusters $\epsilon$ to $0.01$ during distillation in order to remove clusters with very low weighting. We additionally randomly down sample the number of unrestricted adversarial examples to $50,000$ to make inference of the GMM more efficient. For correction, we sample finetuning and testing sets consisting of $256$ images each from every failure scenario. This number of samples captures the breadth of possible images in the scenario, so it is appropriate for tuning and evaluation. We use the finetuning set as the set of failure instances $\mathcal{D}_f$. We use the test set as held out data for evaluating classifier performance on the failure scenarios after correction. During sampling, we fix the sample diversity $\tau$ to $0.5$ for MNIST and $0.01$ for SVHN and German signs because the samples from each of the failure scenarios appear to be in the same class using these values. We finetune over a range of $\lambda$'s in order to find the best balance between training and failure scenario data. We use 3 epochs for MNIST and 5 for both SVHN and German Signs because training converged within both these time frames. During finetuning, we select the model for each $\lambda$ according to the highest training set accuracy for MNIST or validation set accuracy for SVHM and German traffic signs at the end of each finetuning epoch. We select the best model overall as the highest training or validation performance over all $\lambda$'s.

**Annotator Labeling** Because Defuse requires human supervision, we use Amazon Sagemaker Ground Truth to both determine whether clusters generated in the distillation step are failure scenarios and to generate their correct label. In order to determine whether clusters are failure scenarios, we sample 10 instances from each cluster in the distillation step. It is usually apparent the classifier disagrees with many of the ground truth labels within 10 instances, and thus it is appropriate to label the cluster as a failure scenario. For example, in figure 3 it is generally clear the classifier incorrectly predicts the data within only a few examples. As such, 10 instances is a reasonable choice. To reduce noise in the annotation process, we assign the same image to 5 different workers and take the majority annotated label as ground truth. The workers label the images using an interface that includes a single image and the possible labels for that task. We additionally instruct workers to select "None of the above" if the image does not belong to any class and discard these labels. For instance, the MNIST interface includes a single image and buttons for the digits $0 - 9$ along with a

|  | Dataset | # Scenarios | Validation | Test | Failure Scenario |
|---|---|---|---|---|---|
| Before Finetuning | MNIST | - | - | 98.3 | 29.1 |
|  | SVHN | - | 93.6 | 93.2 | 31.2 |
|  | German Signs | - | 98.8 | 98.7 | 27.8 |
| Unrestricted Adversarial Examples | MNIST | - | - | 99.1 | 58.3 |
|  | SVHN | - | 93.1 | 92.9 | 65.4 |
|  | German Signs | - | - | - | - |
| Defuse | MNIST | 19 | - | 99.1 | 96.4 |
|  | SVHN | 6 | 93.0 | 92.8 | 99.9 |
|  | German Signs | 8 | 98.1 | 97.7 | 85.6 |

Figure 4: **Results from the best models** before finetuning, finetuning only on the unrestricted adversarial examples, and finetuning using Defuse. The numbers presented are accuracy on the validation, test set, and failure scenario test set and the absolute number of failure scenarios generated using Defuse. We do not include finetuning on the unrestricted adversarial examples for German Signs because we, the authors, assigned failure scenarios for this data set and thus do not have ground truth labels for individual examples. Critically, the test accuracy on the failure scenarios is high for Defuse indicating that the method successfully corrects the faulty behavior.

"None of the above" button. We provide a screen shot of this interface in figure 14. If more than half (i.e. setting $\rho = 0.5$) of worker labeled instances disagree with the classifier predictions on the 10 instances, we call the cluster a failure scenario. We chose $\rho = 0.5$ because clusters are highly dense with incorrect predictions at this level, making them useful for both understanding model failures and worthwhile for correction. We take the majority prediction over each of the 10 ground truth labels as the label for the failure scenario. As an exception, annotating the German traffic signs data requires specific knowledge of traffic signs. The German traffic signs data ranges across 43 different types of traffic signs. It is not reasonable to assume annotators have enough familiarity with this data and can label it accurately. For this data set, we, the authors, reviewed the distilled clusters and determined which clusters constituted failure scenarios. We labeled the cluster a failure scenario if half the instances appeared to be misclassified.

## 4.2 ILLUSTRATIVE FAILURE SCENARIO EXAMPLES

We demonstrate the potential of Defuse for identifying critical model bugs. We review failure scenarios produced in the three datasets we consider. All together, Defuse produces 19 failure scenarios for MNIST, 6 for SVHN, and 8 for German signs. For each dataset, we provide samples from three failure scenarios in figure 3. The failure scenarios include numerous mislabeled examples. Each failure scenario is composed of mislabeled examples of a similar style. For example, in MNIST, the failure scenario in the upper left hand corner of figure 3 includes a similar style of 4's that are generally predicted incorrectly. The same is true for the failure scenarios in the center and right column where a certain style of 2's and 6's are mistaken. The failure scenarios generally include images which seem difficult to classify. For instance, the misclassified 6's are quite thin making them appear like 1's in some cases. There are similar trends in SVHN and German Signs. In SVHN, particular types of 5's and 8's are misclassified. The same is true in German signs where styles of 50km/h and 30km/h signs are predicted incorrectly. Generally, these methods reveal important bugs in each of the classifiers. It is clear from the MNIST example for instance that very skinny 6's are challenging for the classifier to predict correctly. Further, the German signs classifier has a difficult time with 50km/h signs and tends to frequently mistake them as 80km/h. We provide further samples from other failure scenarios in appendix D. These results clearly demonstrate Defuse reveals insightful model bugs which are useful for model designers to understand.

## 4.3 CORRECTING FAILURE SCENARIOS

We show that Defuse resolves the failure scenarios while maintaining model generalization on the test set. To perform this analysis, we assess accuracy on both the failure scenario test data and test set after correction. It is important for classifier accuracy to improve on the failure scenario data in order

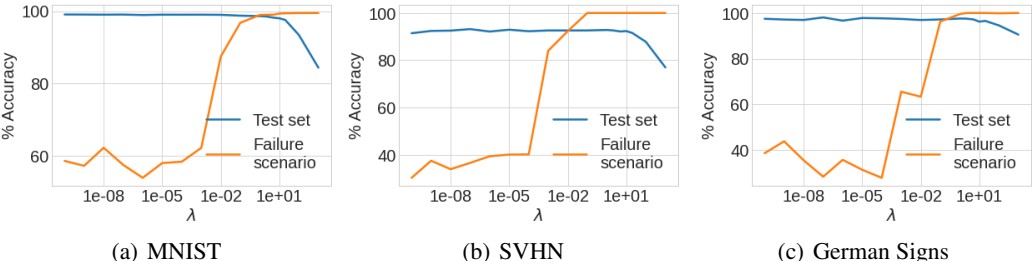

(a) MNIST                    (b) SVHN                    (c) German Signs

Figure 5: **The tradeoff between test set and failure scenario accuracy** running correction. We assess both test set accuracy and accuracy on the test failure scenario data finetuning over a range of $\lambda$'s and plot the trade off. There is an optimal $\lambda$ for each classifier where test set and failure scenario accuracy are both high. This result confirms that the correction step in Defuse adequately balances both generalization and accuracy on the failure scenarios .

to correct the bugs discovered while running Defuse. At the same time, the classifier accuracy on the test set should stay at a similar level or improve indicating that model generalization according to the test set is still strong. We compare Defuse against finetuning only on the unrestricted adversarial examples labeled by annotators. We expect this baseline to be reasonable because related works which focus on robustness to classic adversarial attacks demonstrate that tuning directly on the adversarial examples is effective [Zhang et al. (2019)]. We finetune on the unrestricted adversarial examples sweeping over a range of different $\lambda$'s in the same way as Defuse described in section 4.1. We use this baseline for MNIST and SVHN and not German signs because we, the authors, assigned the failure scenarios for this data set. Thus, we do not have ground truth labels for unrestricted adversarial examples.

We provide an overview of the models before finetuning, finetuning with the unrestricted adversarial examples, and using Defuse in figure 4. Defuse scores highly on the failure scenario data after correction compared to before finetuning. There is only marginal improvement finetuning on the unrestricted adversarial examples. These results indicate Defuse corrects the faulty model performance on the identified failure scenarios. Further, we see the clustering step in Defuse is critical to its success because of the technique's superior performance compared to finetuning on the unrestricted adversarial examples. In addition, there are minor effects on test set performance during finetuning. The test set accuracy increases slightly for MNIST and decreases marginally for SVHN and German Signs for both tuning on the unrestricted adversarial examples and using Defuse. Though the test set performance changes marginally, the increased performance on the failure scenarios demonstrates Defuse's capacity to correct important model errors. Further, we plot the relationship between test set accuracy and failure scenario test accuracy in figure 5. We generally see there is an appropriate $\lambda$ for each model where there is both high test set performance and accuracy on the failure scenarios. All in all, these results indicate Defuse serves as an effective method for correcting specific cases of faulty classifier performance while maintaining model generalization.

### 4.4 ANNOTATOR AGREEMENT

Because we rely on annotators to provide the ground truth labels for the unrestricted adversarial examples, we investigate the agreement between the annotators during labeling. It is important for the annotators to agree on the labels for the unrestricted adversarial examples so that we can have high confidence our evaluation is based on accurately labeled data. We evaluate the annotator agreement through assessing the percent of annotators that voted for the majority label prediction in an unrestricted adversarial example across all the annotated examples. This metric will be high when the annotators are in consensus and low when only a few annotators constitute the majority vote. We provide the annotator agreement on MNIST and SVHN in figure 6 broken down into failure scenario data, non-failure scenario data, and their combination. Interestingly, the failure scenario data has slightly lower annotator agreement indicating these tend to be more ambiguous examples. Further, there is lower agreement on SVHN than MNIST, likely because this data is more complex. All in all, there is generally high annotator agreement across all the data.

| Dataset | Failure Scenario | Non-Failure Scenario | Combined |
|---------|------------------|----------------------|----------|
| MNIST | $78.9.3 \pm 5.4$ | $87.2 \pm 3.2$ | $85.2 \pm 0.1$ |
| SVHN | $66.6 \pm 8.4$ | $83.2 \pm 4.1$ | $82.1 \pm 1.3$ |

Figure 6: **Annotator agreement** on the unrestricted adversarial examples. We plot the mean and standard error of the percent of annotators that voted for the majority label in an unresricted adversarial example across all the annotated examples. We break this down into the failure scenario and non-failure scenario unrestricted adversarial examples and the combination between the two. The annotators are generally in agreement though less so for the failure scenario data, indicating these tend to be more ambiguous examples.

## 5 RELATED WORK

A number of related approaches for improving classifier performance use data created from generative models — mostly generative adversarial networks (GANs) [Sandfort et al. (2019); Milz et al. (2018); Antoniou et al. (2017)]. These methods use GANs to generate instances from classes that are underrepresented in the training data to improve generalization performance. Additional methods use generative models for semi-supervised learning [Kingma et al. (2014); Varma et al. (2016); Kumar et al. (2017); Dumoulin et al. (2016)]. Though these methods are similar in nature to the correction step of our work, a key difference is Defuse focuses on summarizing and presenting high level model failures. Also, [Varma et al. (2017)] provide a system to debug data generated from a GAN when the training set may be inaccurate. Though similar, we ultimately use a generative model to debug a classifier and do not focus on the generative model itself. Last, similar to [Song et al. (2018), Zhao et al. (2018)], [Booth et al. (2020)] provide a method to generate highly confident misclassified instances.

Related to debugging models, [Kang et al. (2018)] focus on model assertions that flag failures during production. Also, [Zhang et al. (2018)] investigate debugging the training set for incorrectly labeled instances. We focus on preemptively identifying model bugs and do not focus on incorrectly labeled test set instances. Additionally, [Ribeiro et al. (2020)] propose a set of behavioral testing tools that help model designers find bugs in NLP models. This technique requires a high level of supervision and thus might not be appropraite in some settings. Last, [Odena et al. (2019)] provide a technique to debug neural networks through perturbing data inputs with various types of noise. By leveraging unrestricted adversarial examples, we distill high level patterns in critical and naturally occurring model bugs. This technique requires minimal human supervision while presenting important types of model errors to designers.

## 6 CONCLUSION

In this paper, we present Defuse: a method that generates and aggregates unrestricted adversarial examples to debug classifiers. Though unrestricted adversarial examples have been proposed in previous works, we harness such examples for the purpose of debugging classifiers. We accomplish this task through identifying failure scenarios: regions in the latent space of a VAE with many unrestricted adversarial examples. On a variety of data sets, we find that samples from failure scenarios are useful in a number of ways. First, failure scenarios are informative for understanding the ways certain models fail. Second, the generative aspect of failure scenarios is very useful for correcting failure scenarios. In our experimental results, we show that these failure scenarios include critical model issues for classifiers with real world impacts — i.e. traffic sign classification — and verify our results using ground truth annotator labels. We demonstrate that Defuse successfully resolves these issues. Although Defuse identifies important errors in classifiers, the technique requires a minimal level of human supervision. Namely, the failure scenarios must be reviewed before correction. In the future, it will be crucial to investigate automatic ways of reviewing failure scenarios.

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

# A   DEFUSE PSUEDO CODE

In algorithm 2, Correct($\cdot$) and Label($\cdot$) are the steps where the annotator decides if the scenario warrants correction and the annotator label for the failure scenario.

---

**Algorithm 1** Identification Step

1: **procedure** IDENTIFY($f, p, q, x, y, a, b$)
2:     $\psi := \{\}$
3:     $\mu, \sigma := q_\phi(x)$
4:     **for** $i \in \{1, ..., Q\}$ **do**
5:         $\epsilon := [\text{Beta}(a, b)_1,$
6:             $..., \text{Beta}(a, b)_M]$
7:         $x_{decoded} := p_\theta(\mu + \epsilon)$
8:         **if** $y \neq f(x_{decoded})$ **then**
9:             $\psi := \psi \cup x_{decoded}$
10:        **end if**
11:    **end for**
12:    Return $\psi$
13: **end procedure**

---

**Algorithm 2** Labeling Step

1: **procedure** LABEL SCENARIOS($Q, \Lambda, p, q, \tau$)
2:     $D_f := \{\}$
3:     **for** $(\mu, \sigma, \pi) \in \Lambda$ **do**
4:         $X_d := \{\}$
5:         **for** $i \in \{1, .., Q\}$ **do**
6:             $X_d := X_d \cup q_\psi(\mathcal{N}(\mu, \tau \cdot \sigma))$
7:         **end for**
8:         **if** Correct($X_d$) **then**
9:             $D_f := D_f \cup \{X_d, \text{Label}(X_d)\}$
10:        **end if**
11:    **end for**
12:    Return $\bigcup D_f$
13: **end procedure**

---

# B   TRAINING DETAILS

## B.1   GMM DETAILS

In all experiments, we use the implementation of Gaussian mixture model with dirichlet process prior from [Pedregosa et al. (2011)]. We run our experiments with the default parameters and full component covariance.

## B.2   MNIST DETAILS

**Model details**   We train a CNN on the MNIST data set using the architecture in figure 7. We used the Adadelta optimizer with the learning rate set to 1. We trained for 5 epochs with a batch size of 64.

| Architecture |
| --- |
| 4x4 conv., 64 ReLU stride 2 |
| 4x4 conv., 64 ReLU stride 2 |
| 4x4 conv., 64 ReLU stride 2 |
| 4x4 conv., 64 ReLU stride 2 |
| Fully connected 256, ReLU |
| Fully connected 256, ReLU |
| Fully connected $10 \times 2$ |

Figure 7: MNIST CNN Architecture

$\beta$-**VAE training details**   We train a $\beta$-VAE on MNIST using the architectures in figure 8 and 9. We set $\beta$ to 4. We trained for 800 epochs using the Adam optimizer with a learning rate of 0.001, a minibatch size of 2048, and $\beta$ set to 0.4. We also applied a linear annealing schedule on the KL-Divergence for 500 optimization steps. We set $z$ to have 10 dimensions.

**Identification**   We performed identification with $Q$ set to 500. We set $a$ and $b$ both to 50. We ran identification over the entire training set. Last, we limited the max allowable size of $\psi$ to 100.

**Distillation**   We ran the distillation step setting $K$, the upper bound on the number of mixtures, to 100. We fixed $\epsilon$ to 0.01 and discarded clusters with mixing proportions less than this value. This left 44 possible scenarios. We set $\tau$ to 0.5 during review. We used Amazon Sagemaker Ground Truth

| Architecture |
| --- |
| 4x4 conv., 32 ReLU stride 2 |
| 4x4 conv., 32 ReLU stride 2 |
| 4x4 conv., 32 ReLU stride 2 |
| Fully connected 256, ReLU |
| Fully connected 256, ReLU |
| Fully connected $15 \times 2$ |

Figure 8: MNIST data set encoder architecture.

| Architecture |
| --- |
| Fully connected 256, ReLU |
| Fully connected 256, ReLU |
| Fully connected 256, ReLU |
| 4x4 transpose conv., 32 ReLU stride 2 |
| 4x4 transpose conv., 32 ReLU stride 2 |
| 4x4 transpose conv., 32 ReLU stride 2 |
| 4x4 transpose conv., 32 Sigmoid stride 2 |

Figure 9: MNIST data set decoder architecture.

to determine failure scenarios and labels. The labeling procedure is described in section 4.1. This produced 19 failure scenarios.

**Correction**   We sampled 256 images from each of the failure scenarios for both finetuning and testing. We finetuned with minibatch size of 256, the Adam optimizer, and learning rate set to 0.001. We swept over a range of correction regularization $\lambda$'s consisting of $[1e-10, 1e-9, 1e-8, 1e-7, 1e-6, 1e-5, 1e-4, 1e-3, 1e-2, 1e-1, 1, 2, 5, 10, 20, 100, 1000]$ and finetuned for 3 epochs on each.

### B.3   German Signs Dataset Details

**Dataset**   The data consists of 26640 training images and 12630 testing images consisting of 43 different types of traffic signs. We randomly split the testing data in half to produce 6315 testing and validation images. Additionally, we resize the images to 128x128 pixels.

**Classifier $f$**   We fine-tuned the ResNet18 model for 20 epochs using Adam with the cross entropy loss, learning rate of 0.001, batch size of 256 on the training data set, and assessed the validation accuracy at the end of each epoch. We saved the model with the highest validation accuracy.

**$\beta$-VAE training details**   We trained for 800 epochs using the Adam optimizer with a learning rate of 0.001, a minibatch size of 2048, and $\beta$ set to 4. We also applied a linear annealing schedule on the KL-Divergence for 500 optimization steps. We set $z$ to have 15 dimensions.

| Architecture |
| --- |
| 4x4 conv., 64 ReLU stride 2 |
| 4x4 conv., 64 ReLU stride 2 |
| 4x4 conv., 64 ReLU stride 2 |
| 4x4 conv., 64 ReLU stride 2 |
| Fully connected 256, ReLU |
| Fully connected 256, ReLU |
| Fully connected $15 \times 2$ |

Figure 10: German signs data set encoder architecture.

**Identification**   We performed identification with $Q$ set to 100. We set $a$ and $b$ both to 75.

| Architecture |
| --- |
| Fully connected 256, ReLU |
| Fully connected 256, ReLU |
| Fully connected 256, ReLU |
| 4x4 transpose conv., 64 ReLU stride 2 |
| 4x4 transpose conv., 64 ReLU stride 2 |
| 4x4 transpose conv., 64 ReLU stride 2 |
| 4x4 transpose conv., 64 ReLU stride 2 |
| 4x4 transpose conv., 64 Sigmoid stride 2 |

Figure 11: German signs data set decoder architecture.

**Distillation**  We ran the distillation step setting $K$ to 100. We fixed $\epsilon$ to 0.01 and discarded clusters with mixing proportions less than this value. This left 38 possible scenarios. We set $\tau$ to 0.01 during review. We determined 8 of these scenarios were particularly concerning.

**Correction**  We finetuned with minibatch size of 256, the Adam optimizer, and learning rate set to 0.001. We swept over a range of correction regularization $\lambda$'s consisting of $[1e-10, 1e-9, 1e-8, 1e-7, 1e-6, 1e-5, 1e-4, 1e-3, 1e-2, 1e-1, 1, 2, 5, 10, 20, 100, 1000]$ and finetuned for 5 epochs on each.

### B.4 SVHN DETAILS

**Dataset**  The data set consists of 73257 training and 26032 testing images. We also randomly split the testing data to create a validation data set. Thus, the final validation and testing set correspond to 13016 images each.

**Classifier $f$**  We fine tuned for 10 epochs using the Adam optimizer, learning rate set to 0.001, and a batch size of 2048. We chose the model which scored the best validation accuracy when measured at the end of each epoch.

**$\beta$-VAE training details**  We trained the $\beta$-VAE for 400 epochs using the Adam optimizer, learning rate 0.001, and minibatch size of 2048. We set $\beta$ to 4 and applied a linear annealing schedule on the Kl-Divergence for 5000 optimization steps. We set $z$ to have 10 dimensions.

| Architecture |
| --- |
| 4x4 conv., 64 ReLU stride 2 |
| 4x4 conv., 64 ReLU stride 2 |
| 4x4 conv., 64 ReLU stride 2 |
| Fully connected 256, ReLU |
| Fully connected 256, ReLU |
| Fully connected $10 \times 2$ |

Figure 12: SVHN data set encoder architecture.

| Architecture |
| --- |
| Fully connected 256, ReLU |
| Fully connected 256, ReLU |
| Fully connected 256, ReLU |
| 4x4 transpose conv., 64 ReLU stride 2 |
| 4x4 transpose conv., 64 ReLU stride 2 |
| 4x4 transpose conv., 64 ReLU stride 2 |
| 4x4 transpose conv., 64 Sigmoid stride 2 |

Figure 13: SVHN data set decoder architecture.

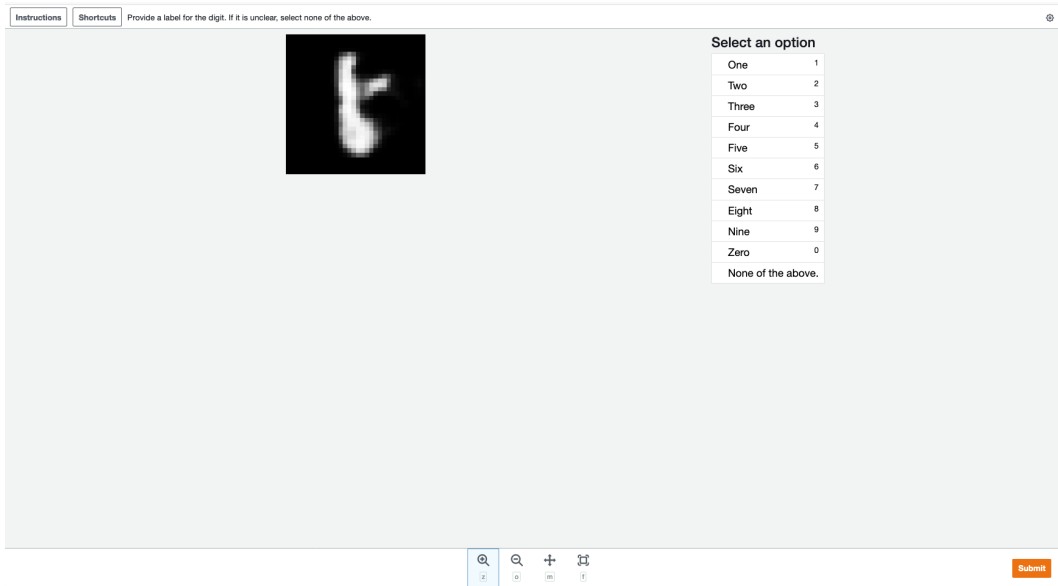

Figure 14: Annotation interface.

**Identification**    We set $Q$ to 100. We also set the maximum size of $\psi$ to 10. We set $a$ and $b$ to 75.

**Distillation**    We set $K$ to 100. We fixed $\epsilon$ to 0.01. The distillation step identified 32 plausible failure scenarios. The annotators deemed 6 of these to be failure scenarios. We set $\tau$ to 0.01 during review.

**Correction**    We set the minibatch size of 2048, the Adam optimizer, and learning rate set to 0.001. We considered a range of $\lambda$'s: $[1e-10, 1e-9, 1e-8, 1e-7, 1e-6, 1e-5, 1e-4, 1e-3, 1e-2, 1e-1, 1, 2, 5, 10, 20, 100, 1000]$. We finetuned for 5 epochs.

### B.5    T-SNE EXAMPLE DETAILS

We run t-SNE on $10,000$ examples from the training data and 516 unrestricted adversarial examples setting perplexity to 30. For the sake of clarity, we do not include outliers from the unrestricted adversarial examples. Namely, we only include unrestricted adversarial examples with $> 1\%$ probability of being in one of the MNIST failure scenario clusters.

## C    ANNOTATOR INTERFACE

We provide a screenshot of the annotator interface in figure 14.

## D    ADDITIONAL EXPERIMENTAL RESULTS

### D.1    ADDITIONAL SAMPLES FROM MNIST FAILURE SCENARIOS

We provide additional examples from 10 randomly selected (no cherry picking) MNIST failure scenarios. We include the annotator consensus label for each failure scenario.

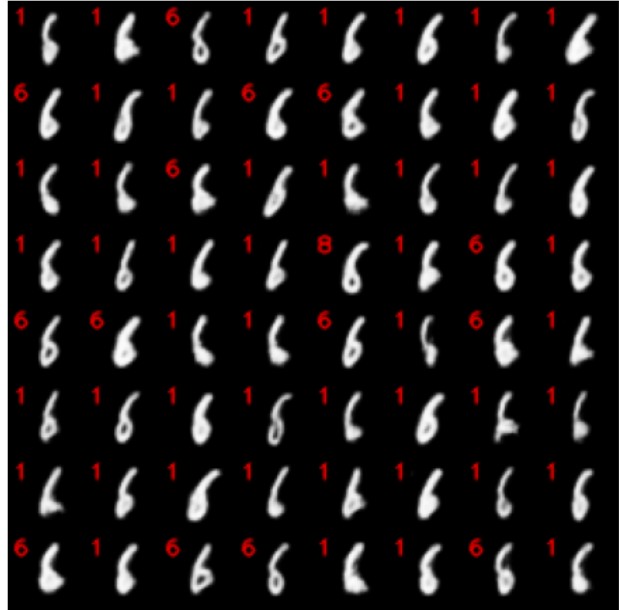

Figure 15: Annotator label 6.

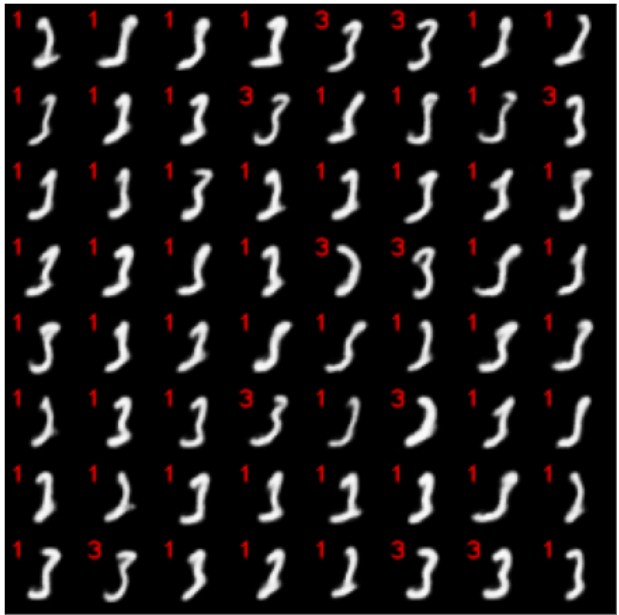

Figure 16: Annotator label 3.

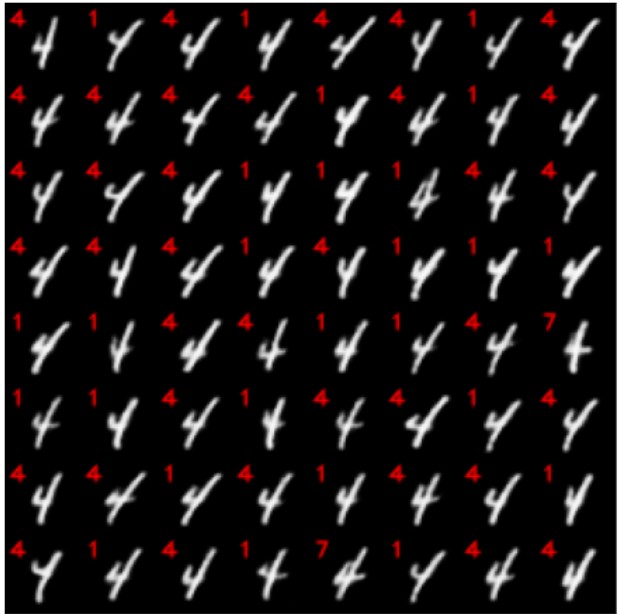

Figure 17: Annotator label 4.

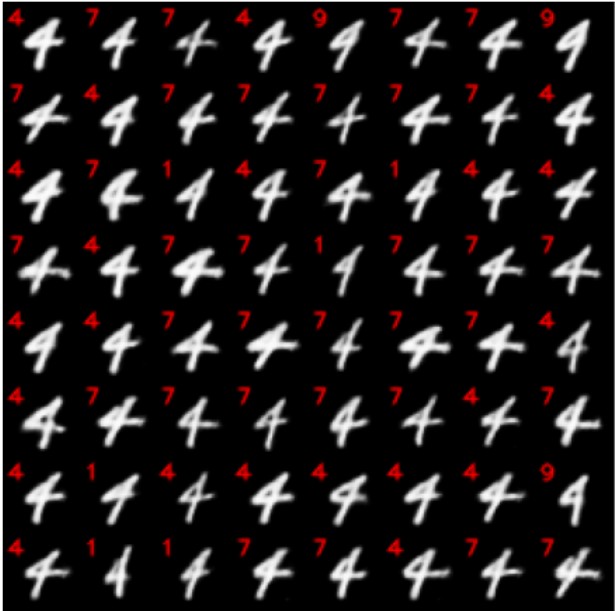

Figure 18: Annotator label 4.

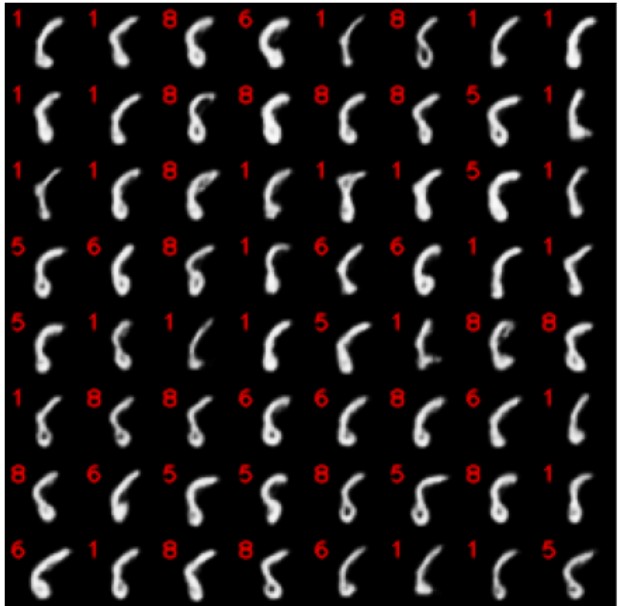

Figure 19: Annotator label 6.

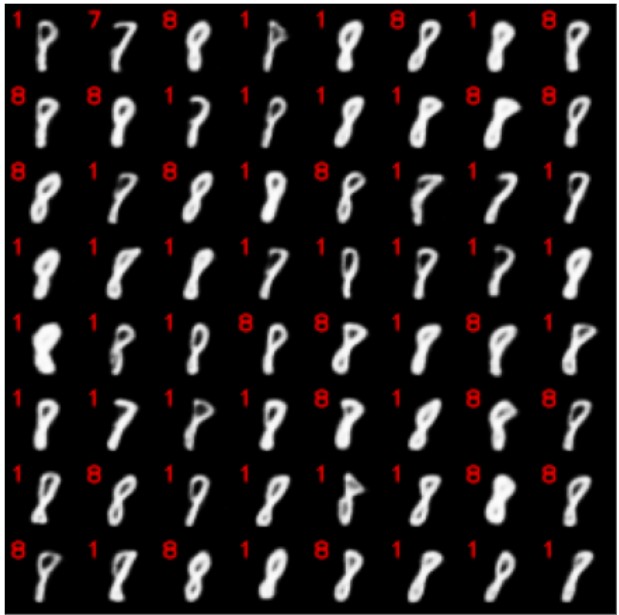

Figure 20: Annotator label 8.

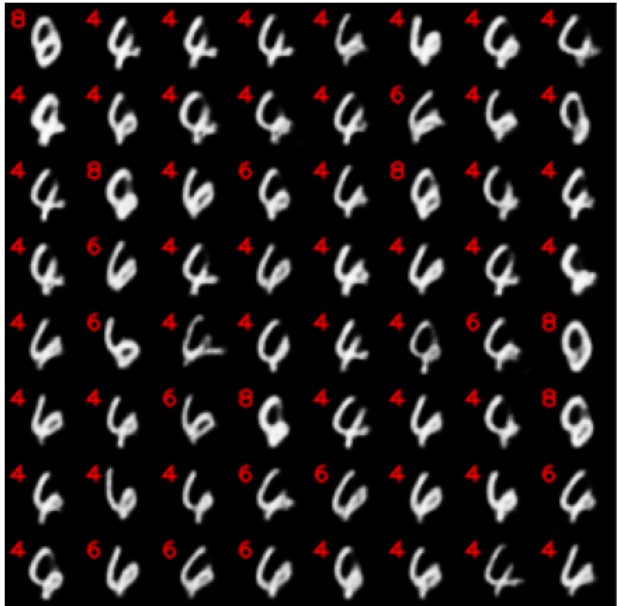

Figure 21: Annotator label 6.

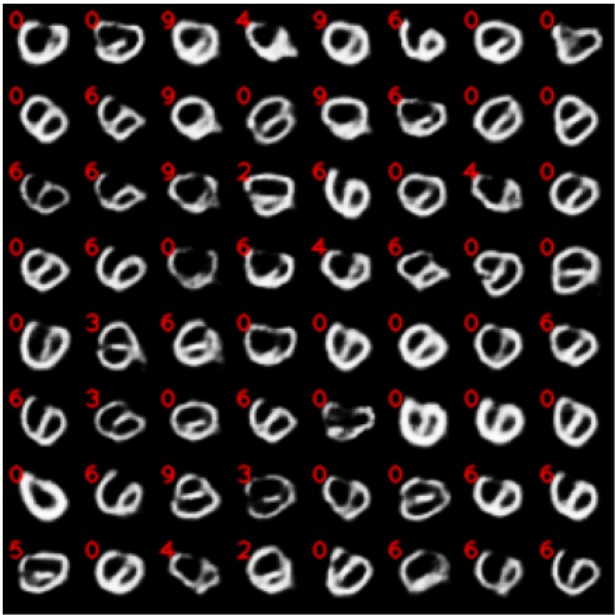

Figure 22: Annotator label 0.

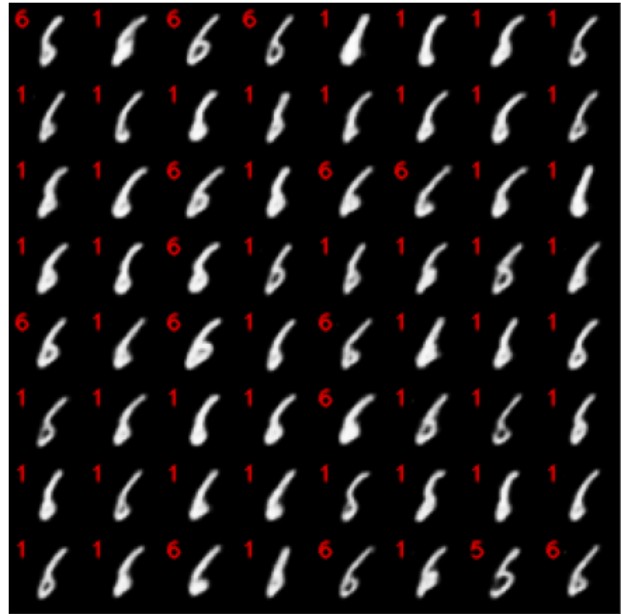

Figure 23: Annotator label 6.

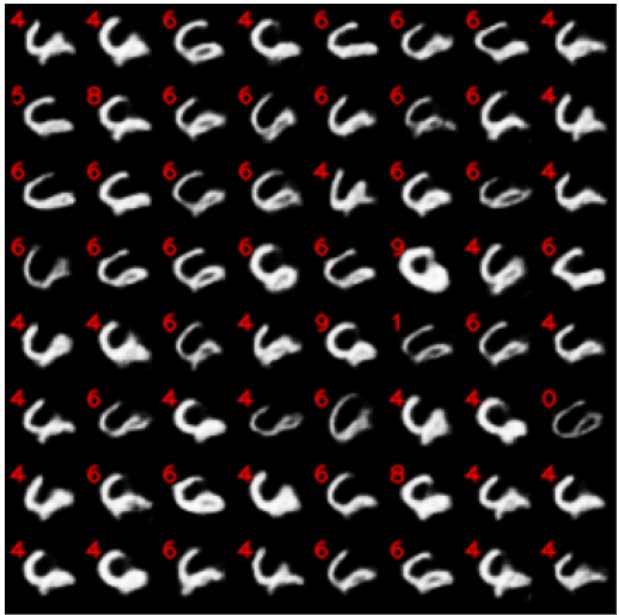

Figure 24: Annotator label 6.

## D.2 ADDITIONAL SAMPLES FROM GERMAN SIGNS FAILURE SCENARIOS

We provide samples from all of the German signs failure scenarios. We provide the names of the class labels in figure 25. For each failure scenario, we indicate our assigned class label in the caption and the classifier predictions in the upper right hand corner of the image.

| ClassId | SignName |
|---------|----------|
| 0 | Speed limit (20km/h) |
| 1 | Speed limit (30km/h) |
| 2 | Speed limit (50km/h) |
| 3 | Speed limit (60km/h) |
| 4 | Speed limit (70km/h) |
| 5 | Speed limit (80km/h) |
| 6 | End of speed limit (80km/h) |
| 7 | Speed limit (100km/h) |
| 8 | Speed limit (120km/h) |
| 9 | No passing |
| 10 | No passing for vehicles over 3.5 metric tons |
| 11 | Right-of-way at the next intersection |
| 12 | Priority road |
| 13 | Yield |
| 14 | Stop |
| 15 | No vehicles |
| 16 | Vehicles over 3.5 metric tons prohibited |
| 17 | No entry |
| 18 | General caution |
| 19 | Dangerous curve to the left |
| 20 | Dangerous curve to the right |
| 21 | Double curve |
| 22 | Bumpy road |
| 23 | Slippery road |
| 24 | Road narrows on the right |
| 25 | Road work |
| 26 | Traffic signals |
| 27 | Pedestrians |
| 28 | Children crossing |
| 29 | Bicycles crossing |
| 30 | Beware of ice/snow |
| 31 | Wild animals crossing |
| 32 | End of all speed and passing limits |
| 33 | Turn right ahead |
| 34 | Turn left ahead |
| 35 | Ahead only |
| 36 | Go straight or right |
| 37 | Go straight or left |
| 38 | Keep right |
| 39 | Keep left |
| 40 | Roundabout mandatory |
| 41 | End of no passing |
| 42 | End of no passing by vehicles over 3.5 metric tons |

Figure 25: German signs class labels.

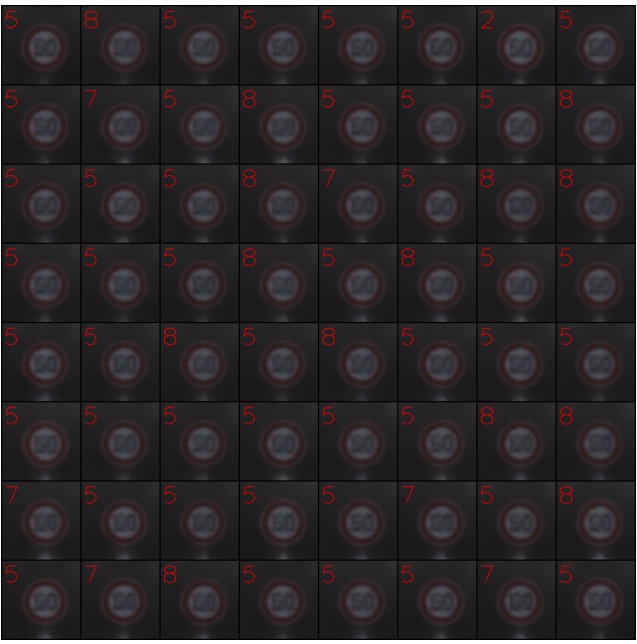

Figure 26: Annotator label 7.

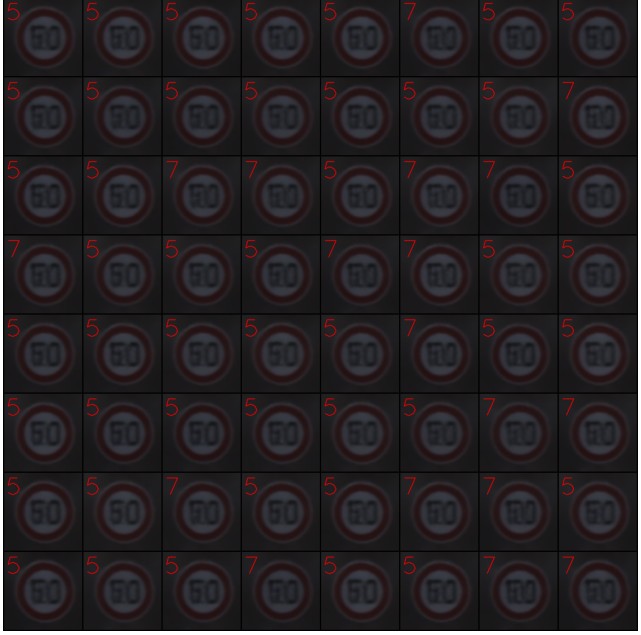

Figure 27: Annotator label 2.

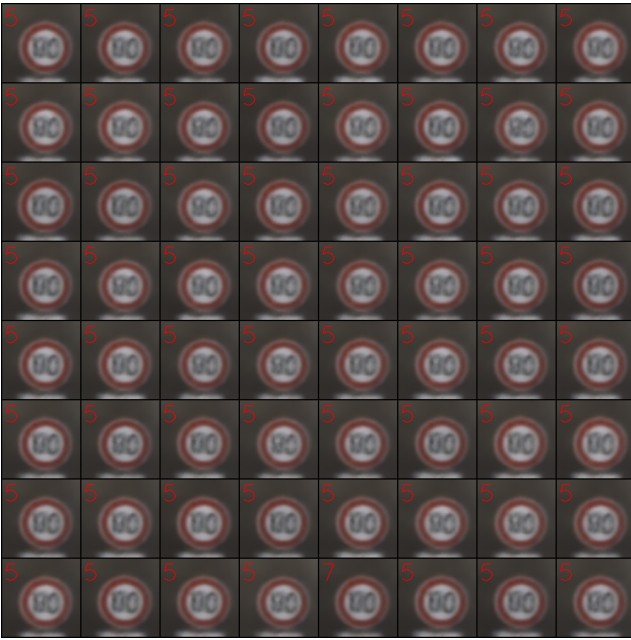

Figure 28: Annotator label 7.

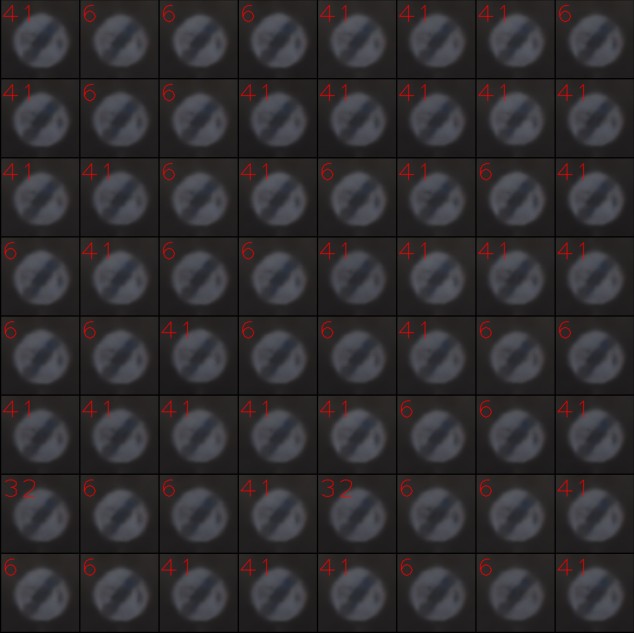

Figure 29: Annotator label 41.

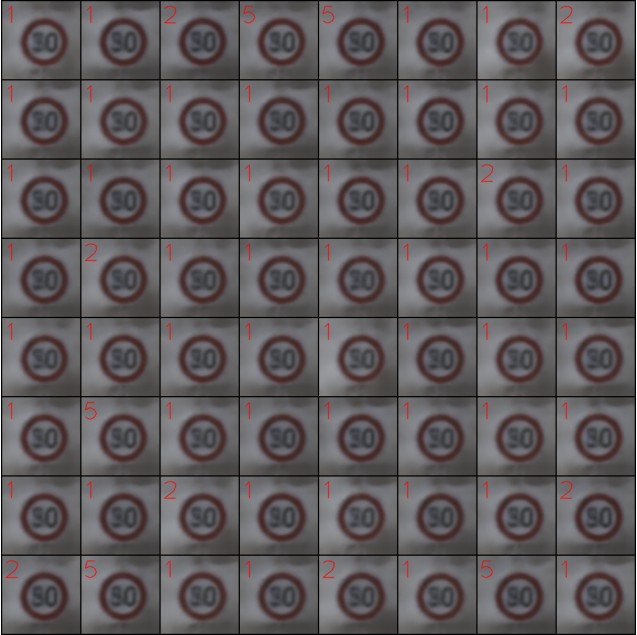

Figure 30: Annotator label 1.

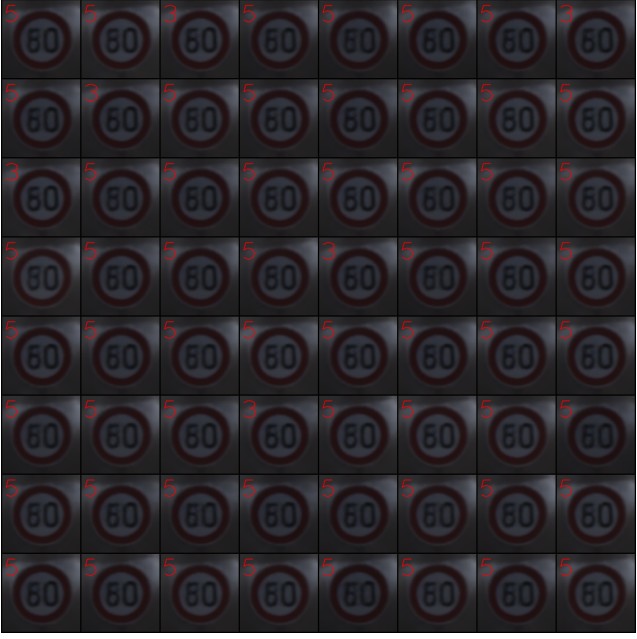

Figure 31: Annotator label 2.

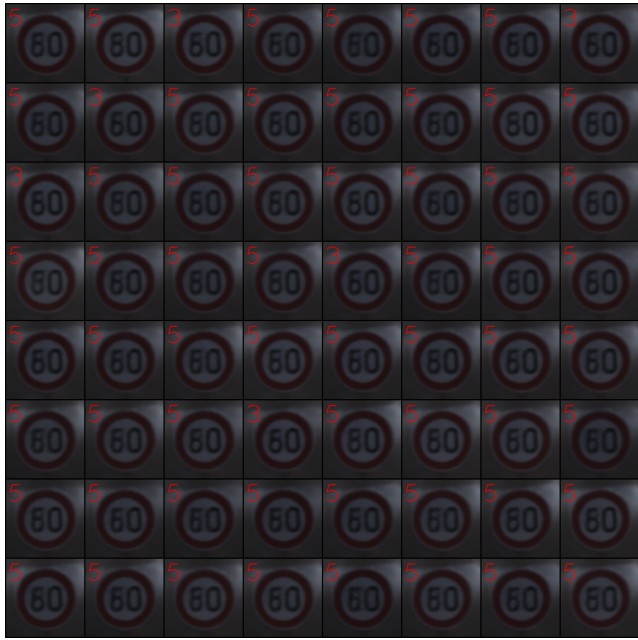

Figure 32: Annotator label 2.

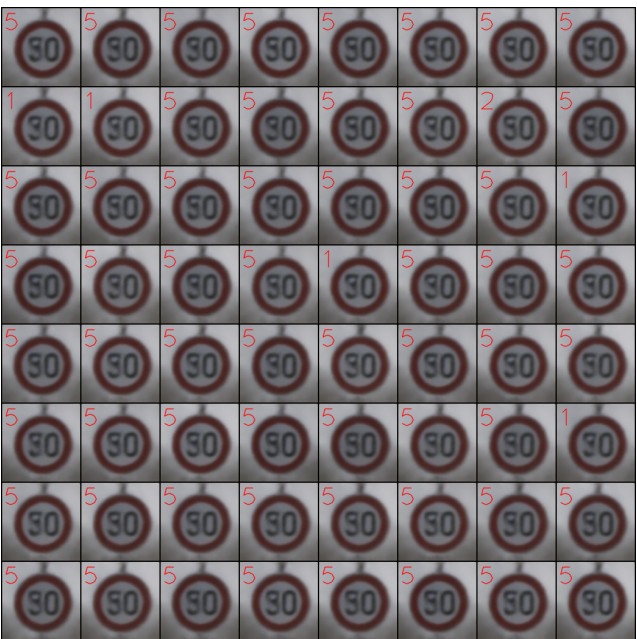

Figure 33: Annotator label 1.

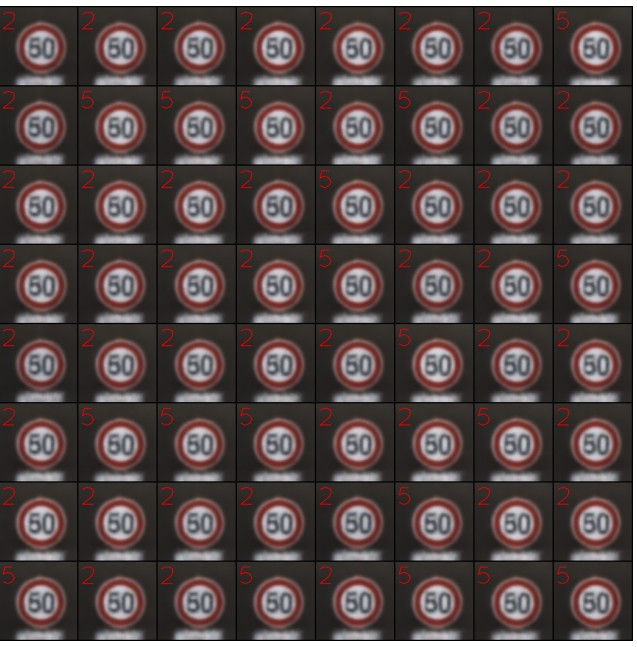

Figure 34: Annotator label 2.

## D.3 ADDITIONAL SAMPLES FROM SVHN FAILURE SCENARIOS

We provide additional samples from each of the SVHN failure scenarios. The digit in the upper left hand corner is the classifier predicted label. The caption includes the Ground Truth worker labels.

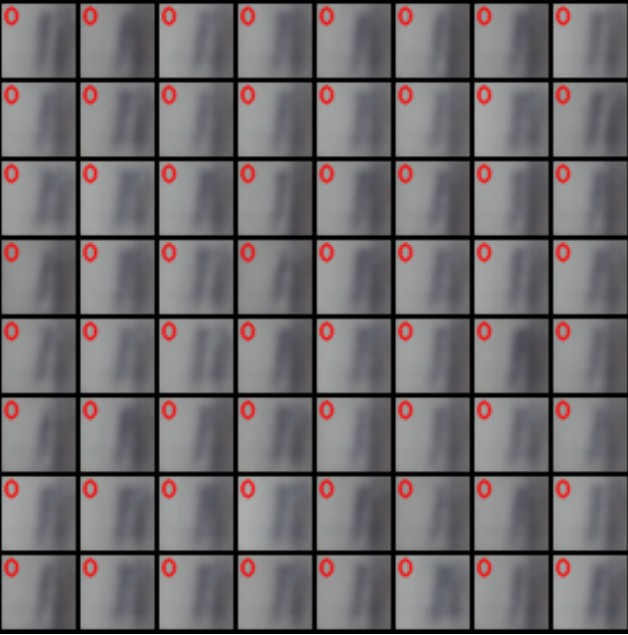

Figure 35: Annotator label 1.

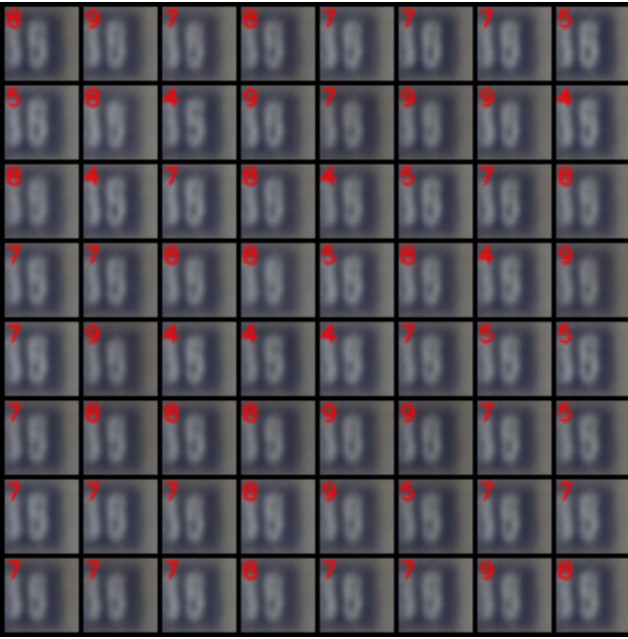

Figure 36: Annotator label 5.

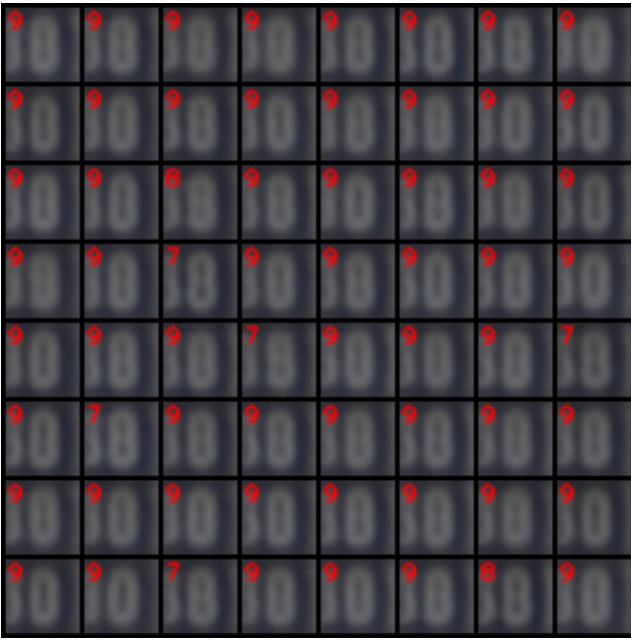

Figure 37: Annotator label 8.

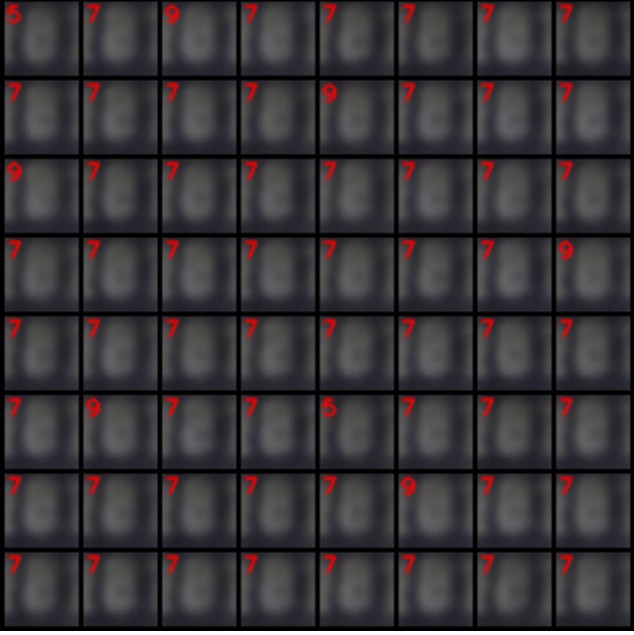

Figure 38: Annotator label 0.

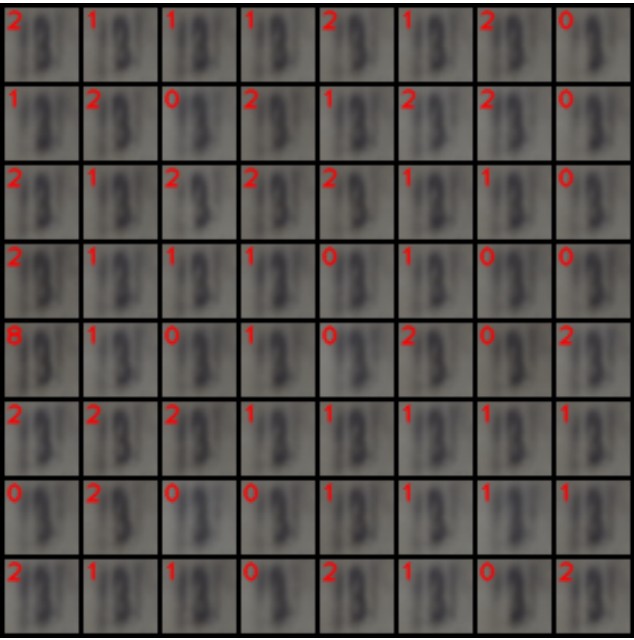

Figure 39: Annotator label 3.

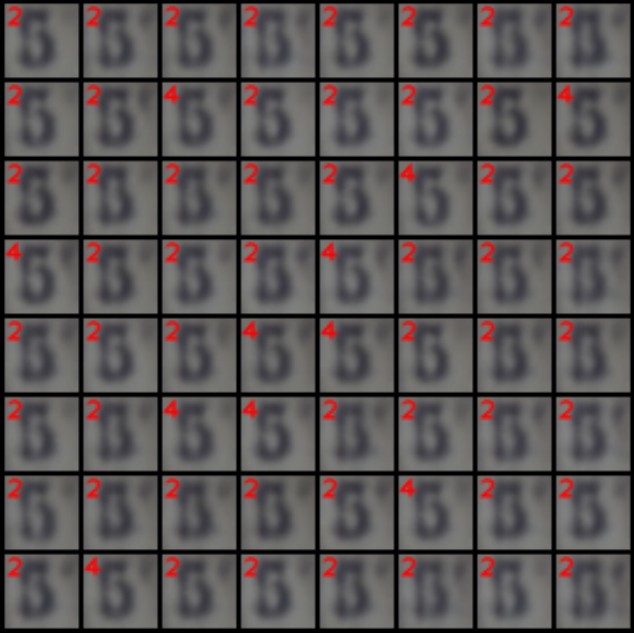

Figure 40: Annotator label 5.

