# OpenReview forum: "Defuse: Debugging Classifiers Through Distilling Unrestricted Adversarial Examples"
_ICLR.cc/2021/Conference — Reject_

### Official Review · AnonReviewer3 · 2020-10-27
**An interesting idea, but experiments and analysis do not support it as a significant contribution**

**Rating:** 4
**Confidence:** 4

**Review:**

The paper proposes a method to identify and correct regions on the data manifold in which a trained classifier fails. The *identification* phase is based on clustering classification failure regions in a GAN latent space and the *correction* phase is based on fine-tuning the classifier with additional synthetic samples from the GAN.

The proposed method is strongly based on Zhao et al 2018 (Generating Natural Adversarial Examples), a method to generate on-manifold black-box adversarial examples using a GAN. The authors of the current paper describe some differences of their identification step from Zhao et al (end of section 3.2.1), but in my opinion they are minor.

The main contribution of the current paper over Zhao et al seems to be clustering the adversarial examples (using GMM) and using them to fine-tune the classifier. This, in my opinion, is potentially an interesting idea, however, the authors do not show sufficient evidence of its success. Specifically, the authors claim to "achieve near perfect failure scenario accuracy with minimal change in test set accuracy", but they do not provide any details (e.g. table of accuracy values on the train, test and adversarial sets before and after the fine-tuning). I would also expect to see an ablation study comparing the proposed method to simply including the adversarial examples found using Zhao et al (w/o GMM fitting and sampling) as additional training example - a standard adversarial defense approach (see e.g. [1]).

Perhaps more importantly, the objective of the proposed method is not, in my opinion, clear. The title and abstract describe the goal as "debugging" a classifier and correcting fail regions, however the described method seems like a defense against on-manifold adversarial attack. If the method, as claimed, helps debugging and correcting the classifier, I would expect to see an improved accuracy on the (natural) unseen test set - not just on the synthetically generated adversarial examples.

The quality and clarity of the writing can be improved as well. A lot of space is allocated to describing well-known methods (e.g. VAE, GMM), however, critical information about the experimental results are missing. I'm also not sure all the formally defined algorithms and equations actually help in the understanding (e.g. algorithm 1, equation 2). Some of the mathematical notations are not standard.

Minor comment: The norm in definition 3.1 is a regular vector norm (l2?) and not a matrix norm.

To summarize:

pros:
- interesting idea (clustering on-manifold failures, labeling them and then using them to improve the classifier)

cons:
- contribution over Zhao et al not well established
- insufficient and inaccurate experimental results
- general quality of writing
- not sure actual work and experiments match the stated objective
- significance

*Update:* Following the authors' response, I upgraded my rating, but I still think there are critical issues with the paper. The most problematic point, in my opinion, is the only-marginal improvement on the test data, indicating that the suggested training method only improves the specific "failure scenarios", making it is similar to adversarial training methods used to gain adversarial robustness. However, the abstract and introduction indicates that the paper helps in debugging in fixing failures in general, which, I think should have been evident in improved test accuracy.

[1] Zhang, Hongyang, et al. "Theoretically principled trade-off between robustness and accuracy." ICML 2019

---

> ### Author Response · Authors · 2020-11-14
> **Response**
>
> We thank the reviewer for their useful comments and interest in the work.  In response to the reviewer’s comments we’ve revised a number of aspects of the paper.
>
> “..but they do not provide any details (e.g. table of accuracy values on the train, test and adversarial sets before and after the fine-tuning)”
>
> - While we previously provided these values in graphical form (what is now figure 5), we provide such a table in figure 4 of the updated paper to more easily parse the results.
>
> “I would also expect to see an ablation study comparing the proposed method to simply including the adversarial examples found using Zhao et al”
>
> - This is a useful point of comparison and thank the reviewer for the suggestion.  We add results finetuning only on the unrestricted adversarial examples.  The results can be found in figure 4.  We find the accuracy on the failure scenario testing data is considerably higher using Defuse than finetuning on the unrestricted adversarial examples.
>
> “Perhaps more importantly, the objective of the proposed method is not, in my opinion, clear.”
>
> - The objective of our work is to systematically find and correct model bugs. Defuse helps to do this through both identifying trends in misclassified data and offers a route to correct the predictions on such data.  See for instance figure 3 in our paper. In the upper right hand corner, a certain style of skinny 6’s are misclassified as 1’s.  This result is insightful for a model designer because it indicates the model struggles with very skinny numbers. Further, our finetuning results demonstrate we can adequate correct the model predictions on these mistakenly classified data indicating Defuse also successfully corrects the fault predictions.
>
> “I would expect to see an improved accuracy on the (natural) unseen test set - not just on the synthetically generated adversarial examples.”
>
> - The test set accuracy marginally improves for MNIST and marginally decreases for SVHN and German signs (figure 4).  We point out however that the important aspect of our work is that accuracy on the failure scenarios (which we have confirmed are model bugs through human verification) are corrected.  We see this is the case with Defuse.
>
> “The quality and clarity of the writing can be improved as well..”
>
> - The reviewer is right to point out there a number of places to improve.  We have reduced the emphasis on VAE + GMM background and added more experimental details.  We have additionally moved the psuedo code for the algorithms to the appendix.
>
> As a minor note, we use VAE’s to perform all our experiments and do not use GANs as the reviewer indicates.  We would appreciate any further response the reviewer has to the above comments and revisions.

---

### Official Review · AnonReviewer4 · 2020-10-29
**The paper describes a technique for debugging classifiers through distilling unrestricted adversarial examples.**

**Rating:** 6
**Confidence:** 3

**Review:**

The technique is described in sufficient detail and the paper is easy to read. Experimental results involving three datasets: MNIST, street view house numbers, and German traffic signs. The experimental results show that the proposed technique finds significant failures in all datasets, including critical failure scenarios. After correction, the performance of the method improves.
An interesting aspect of the method, which distinguishes it from similar techniques, is involvement of users/experts in the training process to indicate the classification errors in order to improve the performance of the method in the future. Engaging users in the training of classifiers has its advantages and disadvantages. For example, it can make easier to create “personalised” classification models that could be applied, e.g. in recommender system or information retrieval, where finding a perfect item depends on user’s subjective perception of certain qualities. At the same time, user involvement in the training process can be tricky if it requires expert judgment as they may not always be available (as the authors demonstrated in the case of their third dataset consisting of German traffic signs). Further, involving user generated assessments requires well defined procedures in terms of requirement of assessors, determining the appropriate number of assessors, resolving disagreements between assessors, to ensure robustness of the final classifier. In the examples provided in the paper, the authors state that they used 5 workers (annotators) and the majority vote was used to decide the final label. What was the inter-annotator agreement? Since using human labellers is a crucial part of the proposed method, I would like to see more discussion of this aspect.

---

> ### Author Response · Authors · 2020-11-14
> **Reponse**
>
> We thank the reviewer for their comments. The reviewer is right to point out that inter annotator agreement is an important aspect to consider.  We add additional details to our annotation process in section 4.1.  Further, we add section 4.4 describing the annotator agreement.  Please see the response to the first reviewer in regards to these details.
>
> We would appreciate any further questions or comments on the new annotator results.

---

### Official Review · AnonReviewer1 · 2020-10-30
**The authors present DEFUSE a system for debugging classifiers using adversarial examples**

**Rating:** 4
**Confidence:** 4

**Review:**

The authors present a system DEFUSE which is geared towards identifying and correcting classifier performance when labels are assigned incorrectly. There are three phases that are used to design DEFUSE: (1) Identify unrestricted adversarial examples using Variational Auto Encoders (2) Use a clustering approach to distill the above examples into failure scenarios and (3) Correct the classifier predictions.

Overall, the idea of using adversarial examples to correct incorrect classifications is very interesting.

The choice of certain algorithms and their parameters needs to be justified clearly. While it is understandable that a non-parametric model be used for the clustering step, it it not clear why a dirichlet process is the best fit. How does this choice compare with other clustering approaches? Do the results generalize?

The paper should be rewritten to have sufficient details of experiments in the text rather than delegating them to the Appendix A.

The motivation of why certain parameters are chosen for experiments should be discussed. For example, "we sample 10 instances from each cluster in the distillation step. We ask 5 workers to label the instance" -- Why are these choices appropriate?
Description of the annotation task ought to be more detailed -- "labeling them ourselves" -- Who constitutes "ourselves"? What is the agreement between the annotators?

Minor comments:

1. Section 3.2: how we identity -> how we identify
2. Section 3.2.3: The paragraph ends with "For instance." The sentence needs to be completed and an example provided.
3. Section 4.1: 32x32 should be replaced with 32X32. Similarly 128x128 should be replaced with 128X128

---

> ### Author Response · Authors · 2020-11-14
> **Response**
>
> We thank the reviewer for their response and comments and appreciate the interest in the work. In response to the reviewer’s points, we’ve significantly improved the experimental detail in the paper.
>
> The choice of certain algorithms and their parameters needs to be justified clearly:
>
> - We better justify the use of the Dirichlet process GMM in section 3.2.2. We point out there are two main requirements with our approach.  First, we must be able to infer the number of clusters from the data.  Second, we must be able to sample new instances from each of the clusters.  Both these requirements greatly limit the clustering approaches we can use. We use the Dirichlet process GMM because the dirichlet process nicely models the clustering problem where the number of clusters in unknown ahead of time satisfying our first criteria.  Additionally, we can sample new instances from the clusters satisfying our second criteria.  Though it could be possible to propose other bayesian clustering methods that meet both these criteria, we focus on evaluating Defuse with this particular choice of clustering method and leave evaluating other clustering methods up to future work.
>
> The paper should be rewritten to have sufficient details of experiments in the text rather than delegating them to the Appendix A:
>
> - We’ve moved many of the Defuse details from the appendix to section 4.1.  We additionally add further justification to our parameter choices.
>
> The motivation of why certain parameters are chosen for experiments should be discussed. For example, "we sample 10 instances from each cluster in the distillation step. We ask 5 workers to label the instance" -- Why are these choices appropriate?
>
> - We’ve added additional justification for our annotation choices in section 4.1.  In addition, we’ve better motivated our parameter choices throughout section 4 in general. In response to this specific question, it is usually apparent the classifier disagrees with many of the ground truth labels within 10 instances, and thus, it is appropriate to label the cluster as a failure scenario.  For example, in figure 3 it is generally clear the classifier incorrectly predicts the data within only a few examples. Thus, 10 instances is a reasonable choice.  We ask 5 workers to label the instance in order to reduce the noise in the annotation process.
>
> Description of the annotation task ought to be more detailed -- "labeling them ourselves" -- Who constitutes "ourselves"?
>
> - For MNIST and SVHN, we use annotator labels.  For German signs, we, the authors reviewed and assigned the failure scenarios.  Though this is less rigorous than the MNIST and SVHN experiments, it is still useful to see the classifier bugs exposed with Defuse.
>
> What is the agreement between the annotators?
>
> - We add section 4.4 describing the annotator agreement during labeling.  We generally find the annotators were in agreement about the labels.  For MNIST, the annotators voted for the majority label on average 85.2% of the time and 82.1% for SVHN over all the unrestricted adversarial examples.
>
> We would appreciate any further reviewer comments or questions from the reviewer to the above responses and revisions.

---

### Author Response · Authors · 2020-11-14
**Author response**

We thank all the reviewers for their useful comments.  We have responded individually to the reviewers below and made substantial changes to the paper. In summary we:

1. Include more Defuse experimental details into the paper in section 4.1.  We additionally provide greater justification for our parameter and model choices in section 4.1 and in regards to the GMM in section 3.2.2.
2. Reduce the description of known methods (VAE+GMM) and focus the writing more on our contributions.
3. Provide more samples from the failure scenarios in figure 3 in the main text.  We emphasize one of our main contributions is the identification of failure scenarios.  The failure scenarios are useful because they summarize the unrestricted adversarial examples into key failure trends in the model.  We bring greater emphasis to this in the paper by highlighting more examples.
3. Compare Defuse to fine tuning on the unrestricted adversarial examples as a baseline (per Reviewer 3's recommendation).  We highlight the Defuse finetuning and baseline results in tabular form in figure 4.  We see that Defuse improves the accuracy on the failure scenarios considerably compared to before finetuning and the baseline.
4. Provide analysis of the annotator agreement in section 4.4.  We see the annotators voted for the majority label on average 85.2% and 82.1% of the time for MNIST and SVHN respectively across the annotated unrestricted adversarial examples.

We also note we had an issue with data storage that effected the MNIST experiments.  We thus reran these experiments with minimal change to our final results.

---

### Decision · Program_Chairs · 2021-01-07
**Final Decision**

**Decision:**

Reject

**Comment:**

The manuscript describes a method for identifying and correcting classifier performance when labels are assigned incorrectly. The identification is based on clustering classification failure regions in a VAE latent space and the correction phase is based on fine-tuning the classifier with additional synthetic samples from the VAE.

Reviewers agreed that the manuscript is not ready for publication. The main issue is that the suggested training method is similar to adversarial training methods used to gain adversarial robustness. The method does not help in debugging and fixing failures in general.